# Leveraging the Exact Likelihood of Deep Latent Variable Models

**Pierre-Alexandre Mattei**
Department of Computer Science
IT University of Copenhagen
pima@itu.dk

**Jes Frellsen**
Department of Computer Science
IT University of Copenhagen
jefr@itu.dk

## Abstract

Deep latent variable models (DLVMs) combine the approximation abilities of deep neural networks and the statistical foundations of generative models. Variational methods are commonly used for inference; however, the exact likelihood of these models has been largely overlooked. The purpose of this work is to study the general properties of this quantity and to show how they can be leveraged in practice. We focus on important inferential problems that rely on the likelihood: estimation and missing data imputation. First, we investigate maximum likelihood estimation for DLVMs: in particular, we show that most unconstrained models used for continuous data have an unbounded likelihood function. This problematic behaviour is demonstrated to be a source of mode collapse. We also show how to ensure the existence of maximum likelihood estimates, and draw useful connections with nonparametric mixture models. Finally, we describe an algorithm for missing data imputation using the exact conditional likelihood of a DLVM. On several data sets, our algorithm consistently and significantly outperforms the usual imputation scheme used for DLVMs.

## 1   Introduction

Dimension reduction aims at summarizing multivariate data using a small number of features that constitute a *code*. Earliest attempts rested on linear projections, leading to Hotelling's (1933) *principal component analysis* (PCA) that has been vastly explored and perfected over the last century (Jolliffe and Cadima, 2016). In recent years, the field has been vividly animated by the successes of *latent variable models* that probabilistically use the low-dimensional features to define powerful generative models. Usually, these latent variable models transform the random code into parameters of a simple distribution. Linear mappings were initially considered, giving rise to factor analysis (Bartholomew et al., 2011) and probabilistic principal component analysis (Tipping and Bishop, 1999). In recent years, much work has been done regarding nonlinear mappings parametrised by deep neural networks, following the seminal papers of Rezende et al. (2014) and Kingma and Welling (2014). These models have led to impressive empirical performance in unsupervised or semi-supervised generative modelling of images (Siddharth et al., 2017), molecular structures (Kusner et al., 2017; Gómez-Bombarelli et al., 2018), arithmetic expressions (Kusner et al., 2017), and single-cell gene expression data (Grønbech et al., 2018). This paper is an investigation of the statistical properties of these models, which remain essentially unknown.

### 1.1   Deep latent variable models

In their most common form, *deep latent variable models* (DLVMs) assume that we are in the presence of a data matrix $\mathbf{X} = (\mathbf{x}_1, ..., \mathbf{x}_n)^T \in \mathcal{X}^n$ that we wish to explain using some latent variables $\mathbf{Z} = (\mathbf{z}_1, ..., \mathbf{z}_n)^T \in \mathbb{R}^{n \times d}$. We assume that $(\mathbf{x}_i, \mathbf{z}_i)_{i \leq n}$ are independent and identically distributed

(i.i.d.) random variables driven by the following generative model:

$$\begin{cases} \mathbf{z} \sim p(\mathbf{z}) \\ p_{\boldsymbol{\theta}}(\mathbf{x}|\mathbf{z}) = \Phi(\mathbf{x}|f_{\boldsymbol{\theta}}(\mathbf{z})). \end{cases} \tag{1}$$

The unobserved random vector $\mathbf{z} \in \mathbb{R}^d$ is called the *latent variable* and usually follows marginally a simple distribution $p(\mathbf{z})$ called the *prior distribution*. The dimension $d$ of the latent space is called the *intrinsic dimension*—and is usually smaller than the dimensionality of the data. The collection $(\Phi(\cdot|\boldsymbol{\eta}))_{\boldsymbol{\eta} \in H}$ is a parametric family of densities with respect to a dominating measure (usually the Lebesgue or the counting measure) called the *observation model*. The function $f_{\boldsymbol{\theta}} : \mathbb{R}^d \to H$ is called a *decoder* or a *generative network*, and is parametrised by a (deep) neural network whose weights are stored in $\boldsymbol{\theta} \in \boldsymbol{\Theta}$. The latent structure of these DLVMs leads to the following marginal distribution of the data:

$$p_{\boldsymbol{\theta}}(\mathbf{x}) = \int_{\mathbb{R}^d} p_{\boldsymbol{\theta}}(\mathbf{x}|\mathbf{z})p(\mathbf{z})d\mathbf{z} = \int_{\mathbb{R}^d} \Phi(\mathbf{x}|f_{\boldsymbol{\theta}}(\mathbf{z}))p(\mathbf{z})d\mathbf{z}. \tag{2}$$

This parametrisation allows to leverage recent advances in deep architectures, such as deep residual networks (Kingma et al., 2016), recurrent networks (Bowman et al., 2016; Gómez-Bombarelli et al., 2018), or batch normalisation (Sønderby et al., 2016).

Several observation models have been considered: in case of discrete multivariate data, products of Bernoulli (or multinomial) distributions; multivariate Gaussian distributions for continuous data; products of Poisson distributions for multivariate count data (Grønbech et al., 2018). Several specific proposals for image data have been made, like the discretised logistic mixture of Salimans et al., 2017). Dirac observation models correspond to *deterministic decoders*, that are used e.g. within generative adversarial networks (Goodfellow et al., 2014), or non-volume preserving transformations (Dinh et al., 2017). Introduced by both Kingma and Welling (2014) and Rezende et al. (2014), the Gaussian and Bernoulli families are the most widely studied, and will be the focus of this article.

## 1.2 Scalable learning through amortised variational inference

The log-likelihood function of a DLVM is, for all $\boldsymbol{\theta} \in \boldsymbol{\Theta}$,

$$\ell(\boldsymbol{\theta}) = \log p_{\boldsymbol{\theta}}(\mathbf{X}) = \sum_{i=1}^{n} \log p_{\boldsymbol{\theta}}(\mathbf{x}_i), \tag{3}$$

which is an extremely challenging quantity to compute that involves potentially high-dimensional integrals. Estimating $\boldsymbol{\theta}$ by maximum likelihood appears therefore out of reach. Consequently, following Rezende et al. (2014) and Kingma and Welling (2014), inference in DLVMs is usually performed using *amortised variational inference*. Variational inference approximatively maximises the log-likelihood by maximising a lower bound known as the *evidence lower bound* (ELBO, see e.g. Blei et al., 2017):

$$\text{ELBO}(\boldsymbol{\theta}, q) = \mathbb{E}_{\mathbf{Z} \sim q} \left[ \log \frac{p(\mathbf{X}, \mathbf{Z})}{q(\mathbf{Z})} \right] = \ell(\boldsymbol{\theta}) - \text{KL}(q||p(\cdot|\mathbf{X})) \leq \ell(\boldsymbol{\theta}), \tag{4}$$

where the *variational distribution* $q$ is a distribution over the space of codes $\mathbb{R}^{n \times d}$. The variational distribution plays the role of a tractable approximation of the posterior distribution of the codes; when this approximation is perfectly accurate, the ELBO is equal to the log-likelihood. *Amortised inference* builds $q$ using a neural network called the *inference network* $g_{\boldsymbol{\gamma}} : \mathcal{X} \to K$, whose weights are stored in $\boldsymbol{\gamma} \in \boldsymbol{\Gamma}$:

$$q_{\boldsymbol{\gamma}, \mathbf{X}}(\mathbf{Z}) = q_{\boldsymbol{\gamma}, \mathbf{X}}(\mathbf{z}_1, ..., \mathbf{z}_n) = \prod_{i=1}^{n} \Psi(\mathbf{z}_i | g_{\boldsymbol{\gamma}}(\mathbf{x}_i)), \tag{5}$$

where $(\Psi(\cdot|\boldsymbol{\kappa}))_{\boldsymbol{\kappa} \in K}$ is a parametric family of distributions over $\mathbb{R}^d$—such as Gaussians with diagonal covariances (Kingma and Welling, 2014). Other kinds of families—built using e.g. normalising flows (Rezende and Mohamed, 2015; Kingma et al., 2016), auxiliary variables (Maaløe et al., 2016; Ranganath et al., 2016), or importance weights (Burda et al., 2016; Cremer et al., 2017)—have been considered for amortised inference, but they will not be central focus of in this paper. Variational

inference for DLVMs then solves the optimisation problem $\max_{\boldsymbol{\theta} \in \boldsymbol{\Theta}, \boldsymbol{\gamma} \in \boldsymbol{\Gamma}} \text{ELBO}(\boldsymbol{\theta}, q_{\boldsymbol{\gamma}}, \mathbf{x})$ using variants of stochastic gradient ascent (see e.g. Roeder et al., 2017, for strategies for computing gradients estimates of the ELBO).

As emphasised by Kingma and Welling (2014), the ELBO resembles the objective function of a popular deep learning model called an *autoencoder* (see e.g. Goodfellow et al., 2016, Chapter 14). This motivates the popular denomination of *encoder* for the inference network $g_{\boldsymbol{\gamma}}$ and *variational autoencoder* (VAE) for the combination of a DLVM with amortised variational inference.

**Contributions.** In this work, we revisit DLVMs by asking: *Is it possible to leverage the properties of $p_{\boldsymbol{\theta}}(\mathbf{x})$ to understand and improve deep generative modelling?* Our main contributions are:

- We show that *maximum likelihood is ill-posed for continuous DLVMs and well-posed for discrete ones*. We link this undesirable property of continuous DLVMs to the mode collapse phenomenon, and illustrate it on a real data set.

- We draw a connection between DLVMs and nonparametric statistics, and show that *DLVMs can be seen as parsimonious submodels of nonparametric mixture models*.

- We leverage this connection to provide a way of finding an *upper bound of the likelihood based on finite mixtures*. Combined with the ELBO, this bound allows us to provide useful "sandwichings" of the exact likelihood. We also prove that this bound characterises the large capacity behaviour of DLVMs.

- When dealing with missing data, we show how a simple modification of an approximate scheme proposed by Rezende et al. (2014) allows us to *draw according to the exact conditional distribution of the missing data*. On several data sets and missing data scenarios, our algorithm consistently outperforms the one of Rezende et al. (2014), while having the same computational cost.

## 2 Is maximum likelihood well-defined for deep latent variable models?

In this section, we investigate the properties of maximum likelihood estimation for DLVMs with Gaussian and Bernoulli observation models.

### 2.1 On the boundedness of the likelihood of deep latent variable models

Deep generative models with Gaussian observation models assume that the data space is $\mathcal{X} = \mathbb{R}^p$, and that the observation model is the family of $p$-variate full-rank Gaussian distributions. The conditional distribution of each data point is consequently

$$p_{\boldsymbol{\theta}}(\mathbf{x}|\mathbf{z}) = \mathcal{N}(\mathbf{x}|\boldsymbol{\mu}_{\boldsymbol{\theta}}(\mathbf{z}), \boldsymbol{\Sigma}_{\boldsymbol{\theta}}(\mathbf{z})), \tag{6}$$

where $\boldsymbol{\mu}_{\boldsymbol{\theta}} : \mathbb{R}^d \to \mathbb{R}^p$ and $\boldsymbol{\Sigma}_{\boldsymbol{\theta}} : \mathbb{R}^d \to \mathcal{S}_p^{++}$ are two continuous functions parametrised by neural networks whose weights are stored in a parameter $\boldsymbol{\theta}$. These two functions constitute the decoder of the model. This leads to the log-likelihood

$$\ell(\boldsymbol{\theta}) = \sum_{i=1}^{n} \log \left( \int_{\mathbb{R}^d} \mathcal{N}(\mathbf{x}_i|\boldsymbol{\mu}_{\boldsymbol{\theta}}(\mathbf{z}), \boldsymbol{\Sigma}_{\boldsymbol{\theta}}(\mathbf{z})) p(\mathbf{z}) d\mathbf{z} \right). \tag{7}$$

This model can be seen as a special case of *infinite mixture of Gaussian distributions*. However, it is well-known that maximum likelihood is ill-posed for *finite* Gaussian mixtures (see e.g. Le Cam, 1990). Here, by "ill-posed", we mean that, inside the parameter space, there exists no maximiser of the likelihood function, which corresponds to the first condition given by Tikhonov and Arsenin (1977, p.7 ). This happens because the likelihood function is unbounded above. Moreover, the infinite maxima of the likelihood happen to be very poor generative models, whose density collapse around some of the data points. This problematic behaviour of a model quite similar to DLVMs motivates the question: *is the likelihood function of DLVMs bounded above?*

In this section, we will not make any particular parametric assumption about the prior distribution of the latent variable $\mathbf{z}$. While Kingma and Welling (2014) and Rezende et al. (2014) originally proposed to use isotropic Gaussian distributions, more complex learnable priors have also been proposed (e.g. Tomczak and Welling, 2018). We simply make the natural assumptions that $\mathbf{z}$ is continuous and has zero mean. Many different neural architectures have been explored regarding

the parametrisation of the decoder. For example, Kingma and Welling (2014) consider multilayer perceptrons (MLPs) of the form

$$\boldsymbol{\mu}_{\boldsymbol{\theta}}(\mathbf{z}) = \mathbf{V} \tanh{(\mathbf{W}\mathbf{z} + \mathbf{a})} + \mathbf{b}, \ \boldsymbol{\Sigma}_{\boldsymbol{\theta}}(\mathbf{z}) = \mathrm{Diag}\left(\exp\left(\boldsymbol{\alpha}\tanh{(\mathbf{W}\mathbf{z} + \mathbf{a})} + \boldsymbol{\beta}\right)\right), \quad (8)$$

where $\boldsymbol{\theta} = (\mathbf{W}, \mathbf{a}, \mathbf{V}, \mathbf{b}, \boldsymbol{\alpha}, \beta)$. The weights of the decoder are $\mathbf{W} \in \mathbb{R}^{h \times d}, \mathbf{a} \in \mathbb{R}^h, \mathbf{V}, \boldsymbol{\alpha} \in \mathbb{R}^{p \times h}$, and $\mathbf{b}, \boldsymbol{\beta} \in \mathbb{R}^p$. The integer $h \in \mathbb{N}^*$ is the (common) number of *hidden units* of the MLPs. Much more complex parametrisations exist, but we will see that this one, arguably one of the most rigid, is already too flexible for maximum likelihood. Actually, we will show that an even much less flexible family of MLPs with a single hidden unit is problematic and leads the model to collapse around a data point. Let $\mathbf{w} \in \mathbb{R}^p$ and let $(\alpha_k)_{k \geq 1}$ be a sequence of nonnegative real numbers such that $\alpha_k \to +\infty$ as $k \to +\infty$. Let us consider $i^* \in \{1, ..., n\}$: this arbitrary index will represent the observation around which the model will collapse. Using the parametrisation (8), we consider the sequences of parameters $\boldsymbol{\theta}_k^{(i^*, \mathbf{w})} = (\alpha_k \mathbf{w}^T, 0, \mathbf{0}_p, \mathbf{x}_{i^*}, \alpha_k \mathbf{1}_p, -\alpha_k \mathbf{1}_p)$. This leads to the simplified decoders:

$$\boldsymbol{\mu}_{\boldsymbol{\theta}_k^{(i^*, \mathbf{w})}}(\mathbf{z}) = \mathbf{x}_{i^*}, \ \boldsymbol{\Sigma}_{\boldsymbol{\theta}_k^{(i^*, \mathbf{w})}}(\mathbf{z}) = \exp\left(\alpha_k \tanh\left(\alpha_k \mathbf{w}^T \mathbf{z}\right) - \alpha_k\right) \mathbf{I}_p. \quad (9)$$

As shown by next theorem, these sequences of decoders lead to the divergence of the log-likelihood function.

**Theorem 1.** *For all $i^* \in \{1, ..., n\}$ and $\mathbf{w} \in \mathbb{R}^d \setminus \{0\}$, we have $\lim_{k \to +\infty} \ell\left(\boldsymbol{\theta}_k^{(i^*, \mathbf{w})}\right) = +\infty$.*

A detailed proof is provided in Appendix A (all appendices of this paper are available as supplementary material). Its cornerstone is the fact that the sequence of functions $\boldsymbol{\Sigma}_{\boldsymbol{\theta}_k^{(i^*, \mathbf{w})}}$ converges to a function that outputs both singular and nonsingular covariances, leading to the explosion of $\log p_{\boldsymbol{\theta}_k^{(i^*, \mathbf{w})}}(\mathbf{x}_{i^*})$ while all other terms of the log-likelihood remain bounded below by a constant.

Using simple MLP-based parametrisations such a the one of Kingma and Welling (2014) therefore brings about an unbounded log-likelihood function. A natural question that follows is: do these infinite suprema lead to useful generative models? The answer is no. Actually, none of the functions considered in Theorem 1 are particularly useful, because of the use of a constant mean function. This is formalised in the next proposition, that exhibits a strong link between likelihood blow-up and the *mode collapse* phenomenon.

**Proposition 1.** *For all $k \in \mathbb{N}^*$, $i^* \in \{1, ..., n\}$, and $\mathbf{w} \in \mathbb{R}^d \setminus \{0\}$, the distribution $p_{\boldsymbol{\theta}_k^{(i^*, \mathbf{w})}}$ is spherically symmetric and unimodal around $\mathbf{x}_{i^*}$.*

A proof is provided in Appendix B. This is a direct consequence of the constant mean function.

The spherical symmetry implies that the distribution of these "optimal" deep generative model will lead to uncorrelated variables, and the unimodality will lead to poor sample diversity. This behaviour is symptomatic of mode collapse, which remains one of the most challenging drawbacks of generative modelling (Arora et al., 2018). While mode collapse has been extensively investigated for adversarial training (e.g. Arora et al., 2018; Lucas et al., 2018), this phenomenon is also known to affect VAEs (Richardson and Weiss, 2018).

Unregularised gradient-based optimisation of a tight lower bound of this unbounded likelihood is therefore likely to follow these (uncountably many) paths to blow-up. This gives a theoretical foundation to the necessary regularisation of VAEs that was already noted by Rezende et al. (2014) and Kingma and Welling (2014). For example, using weight decay as in Kingma and Welling (2014) is likely to help avoiding these infinite maxima. This difficulty to learn the variance was also experimentally noticed by Takahashi et al. (2018), and may explain the choice made by several authors to use a constant variance function $\boldsymbol{\Sigma}(\mathbf{z}) = \sigma_0 \mathbf{I}_p$, where $\sigma_0$ can be either fixed (Zhao et al., 2017) or learned via approximate maximum likelihood (Pu et al., 2016). Dai et al. (2018) independently showed that the VAE objective is also unbounded in the case where such a constant variance function is combined with a nonparametric mean function. An interesting feature of our result is that it only involves a decoder of very low capacity.

**Tackling the unboundedness of the likelihood.** Let us go back to a parametrisation which is not necessarily MLP-based. Even in this general context, it is possible to tackle the unboundedness of the likelihood using additional constraints on $\boldsymbol{\Sigma}_{\boldsymbol{\theta}}$. Specifically, for each $\xi \geq 0$, we will consider the set $\mathcal{S}_p^\xi = \{\mathbf{A} \in \mathcal{S}_p^+ | \min(\mathrm{Sp}\mathbf{A}) \geq \xi\}$, where, for all $\mathbf{A} \in \mathcal{S}_p^+$, $\mathrm{Sp}\mathbf{A}$ denotes the spectrum of $\mathbf{A}$. Note that $\mathcal{S}_p^0 = \mathcal{S}_p^+$. This simple spectral constraint allows to end up with a bounded likelihood.

**Proposition 2.** *Let $\xi > 0$. If the parametrisation of the decoder is such that the image of $\boldsymbol{\Sigma_\theta}$ is included in $\mathcal{S}_p^\xi$ for all $\boldsymbol{\theta}$, then the log-likelihood function is upper bounded by $-np \log \sqrt{2\pi\xi}$.*

*Proof.* For all $i \in \{1, ..., n\}$, we have $p(\mathbf{x}_i|\boldsymbol{\mu_\theta}, \boldsymbol{\Sigma_\theta}) \leq (2\pi\xi)^{-2p/2}$, using the fact that the determinant of $\boldsymbol{\Sigma_\theta}(\mathbf{z})$ is lower bounded by $\xi^p$ for all $\mathbf{z} \in \mathbb{R}^d$ and that the exponential of a negative number is smaller than one. Therefore, the likelihood function is bounded above by $1/(2\pi\xi)^{np/2}$.  $\square$

Similar constraints have been proposed to solve the ill-posedness of maximum likelihood for finite Gaussian mixtures (e.g. Hathaway, 1985; Biernacki and Castellan, 2011). In practice, implementing such constraints can be easily done by adding a constant diagonal matrix to the output of the covariance decoder.

**What about other parametrisations?**    We chose a specific and natural parametrisation in order to obtain a constructive proof of the unboundedness of the likelihood. However, virtually any other deep parametrisation that does not include covariance constraints will be affected by our result, because of the universal approximation abilities of neural networks (see e.g. Goodfellow et al., 2016, Section 6.4.1).

**Bernoulli DLVMs do not suffer from unbounded likelihood.**    When $\mathcal{X} = \{0, 1\}^p$, Bernoulli DLVMs assume that $(\Phi(\cdot|\boldsymbol{\eta}))_{\boldsymbol{\eta} \in H}$ is the family of $p$-variate multivariate Bernoulli distributions (i.e. the family of products of $p$ univariate Bernoulli distributions). In this case, maximum likelihood is well-posed.

**Proposition 3.** *Given any possible parametrisation, the log-likelihood function of a deep latent model with a Bernoulli observation model is everywhere negative.*

*Proof.* This directly follows from the fact that the Bernoulli density is always smaller than one.  $\square$

## 2.2   Towards data-dependent likelihood upper bounds

We have determined under which conditions maximum likelihood estimates exist, and have computed simple upper bounds on the likelihood functions. Since they do not depend on the data, these bounds are likely to be very loose. A natural follow-up issue is to seek tighter, data-dependent upper bounds that remain easily computable. Such bounds are desirable because, combined with ELBOs, they would allow sandwiching the likelihood between two bounds.

To study this problem, let us take a step backwards and consider a more general infinite mixture model. Precisely, given any distribution $G$ over the generic parameter space $H$, we define the *nonparametric mixture model* (see e.g. Lindsay, 1995, Chapter 1) as:

$$p_G(\mathbf{x}) = \int_H \Phi(\mathbf{x}|\boldsymbol{\eta}) dG(\boldsymbol{\eta}). \tag{10}$$

Note that there are many ways for a mixture model to be nonparametric (e.g. having some nonparametric components, an infinite but countable number of components, or an uncountable number of components). In this case, this comes from the fact that the model parameter is the *mixing distribution* $G$, which belongs to the set $\mathcal{P}$ of all probability measures over $H$. The log-likelihood of any $G \in \mathcal{P}$ is given by $\ell(G) = \sum_{i=1}^{n} \log p_G(\mathbf{x}_i)$.

When $G$ has a finite support of cardinal $k \in \mathbb{N}^*$, $p_G$ is a finite mixture model with $k$ components. When the mixing distribution $G$ is generatively defined by the distribution of a random variable $\boldsymbol{\eta}$ such that $\mathbf{z} \sim p(\mathbf{z})$, $\boldsymbol{\eta} = f_{\boldsymbol{\theta}}(\mathbf{z})$, we exactly end up with a deep generative model with decoder $f_{\boldsymbol{\theta}}$. Therefore, the nonparametric mixture is a more general model that the DLVM. The fact that the mixing distribution of a DLVM is intrinsically low-dimensional leads us to interpret the DLVM as a *parsimonious submodel of the nonparametric mixture model*. This also gives us an immediate upper bound on the likelihood of any decoder $f_{\boldsymbol{\theta}}$: $\ell(\boldsymbol{\theta}) \leq \max_{G \in \mathcal{P}} \ell(G)$.

Of course, in many cases, this upper bound will be infinite (for example in the case of an unconstrained Gaussian observation model). However, under the conditions of boundedness of the likelihood of deep Gaussian models, the bound is finite and attained for a finite mixture model with no more components than data points.

**Theorem 2.** *Assume that $(\Phi(\cdot|\boldsymbol{\eta}))_{\boldsymbol{\eta}\in H}$ is the family of multivariate Bernoulli distributions or the family of Gaussian distributions with the spectral constraint of Proposition 2. The likelihood of the corresponding nonparametric mixture model is maximised for a finite mixture model of $k \leq n$ distributions from the family $(\Phi(\cdot|\boldsymbol{\eta}))_{\boldsymbol{\eta}\in H}$.*

A detailed proof is provided in Appendix C. The main tool of the proof of this rather surprising result is Lindsay's (1983) geometric analysis of the likelihood of nonparametric mixtures, based on Minkovski's theorem. Specifically, Lindsay's (1983) Theorem 3.1 ensures that, when the trace of the likelihood curve is compact, the likelihood function is maximised for a finite mixture. For the Bernoulli case, compactness of the curve is immediate; for the Gaussian case, we use a compactification argument inspired by van der Vaart and Wellner (1992).

Assume now that the conditions of Theorem 2 are satisfied. Let us denote a maximum likelihood estimate of $G$ as $\hat{G}$. For all $\boldsymbol{\theta}$, we therefore have

$$\ell(\boldsymbol{\theta}) \leq \ell(\hat{G}), \qquad (11)$$

which gives an *upper bound* on the likelihood. We call the difference $\ell(\hat{G}) - \ell(\boldsymbol{\theta})$ the *parsimony gap* (see Fig. 1). By sandwiching the exact likelihood between this bound and an ELBO, we can also have guarantees on how far a posterior approximation $q$ is from the true posterior:

$$\mathrm{KL}(q||p(\cdot|\mathbf{X})) \leq \ell(\hat{G}) - \mathrm{ELBO}(\boldsymbol{\theta}, q). \qquad (12)$$

Note that finding upper bounds of the likelihood of latent variable models is usually harder than finding lower bounds (Grosse et al., 2015; Dieng

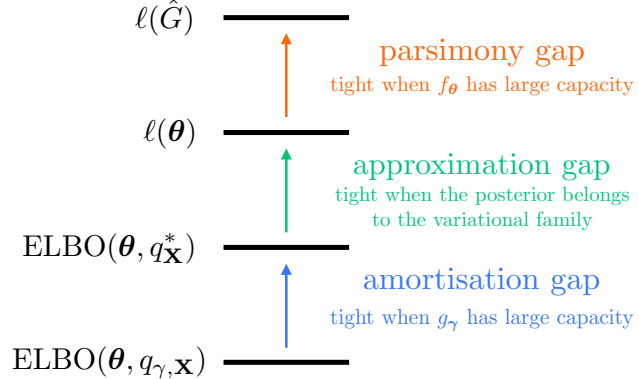

Figure 1: The parsimony gap represents the amount of likelihood lost due to the architecture of the decoder. The approximation gap expresses how far the posterior is from the variational family, and the amortisation gap appears due to the limited capacity of the encoder (Cremer et al., 2018).

et al., 2017). From a computational perspective, the estimate $\hat{G}$ can be found using the expectation-maximisation algorithm for finite mixtures (Dempster et al., 1977)—although it only ensures to find a local optimum. Some strategies guaranteed to find a global optimum have also been developed (e.g. Lindsay, 1995, Chapter 6, or Wang, 2007).

Now that computationally approachable upper bounds have been derived, the question remains whether or not these bounds can be tight. Actually, as shown by next theorem, tightness of the parsimony gap occurs when the decoder has universal approximation abilities. In other words, *the nonparametric upper bound characterises the large capacity limit of the decoder.*

**Theorem 3 (Tightness of the parsimony gap).** *Assume that*

1. *$(\Phi(\cdot|\boldsymbol{\eta}))_{\boldsymbol{\eta}\in H}$ is the family of multivariate Bernoulli distributions or the family of Gaussian distributions with the spectral constraint of Proposition 2.*

2. *The decoder has universal approximation abilities : for any compact $C \subset \mathbb{R}^d$ and continuous function $f : C \to H$, for all $\varepsilon > 0$, there exists $\boldsymbol{\theta}$ such that $||f - f_{\boldsymbol{\theta}}||_\infty < \varepsilon$.*

*Then, for all $\varepsilon > 0$, there exists $\boldsymbol{\theta} \in \boldsymbol{\Theta}$ such that $\ell(\hat{G}) \geq \ell(\boldsymbol{\theta}) \geq \ell(\hat{G}) - \varepsilon$.*

A detailed proof is provided in Appendix D. The main idea is to split the code space into a compact set made of several parts that will represent the mixture components, and an unbounded set of very small prior mass. The universal approximation property is finally used for this compact set.

The universal approximation condition is satisfied for example by MLPs with nonpolynomial activations (Leshno et al., 1993). Combined with the work of Cremer et al. (2018), who studied the large capacity limit of the *encoder*, this result describes the general behaviour of a VAE in the large capacity limit (see Fig. 1). Note eventually that Rezende and Viola (2018) analysed the large capacity behaviour of the VAE objective, and also found connections with finite mixtures.

# 3 Missing data imputation using the exact conditional likelihood

In this section, we assume that a variational autoencoder has been trained, and that some data is missing at test time. The couple decoder/encoder obtained after training is denoted by $f_{\boldsymbol{\theta}}$ and $g_{\boldsymbol{\gamma}}$. Let $\mathbf{x} \in \mathcal{X}$ be a new data point that consists of some observed features $\mathbf{x}^{\mathrm{obs}}$ and missing data $\mathbf{x}^{\mathrm{miss}}$. Since we have a probabilistic model $p_{\boldsymbol{\theta}}$ of the data, an ideal way of imputing $\mathbf{x}^{\mathrm{miss}}$ would be to generate some data according to the *conditional distribution*

$$p_{\boldsymbol{\theta}}(\mathbf{x}^{\mathrm{miss}}|\mathbf{x}^{\mathrm{obs}}) = \int_{\mathbb{R}^d} p_{\boldsymbol{\theta}}(\mathbf{x}^{\mathrm{miss}}|\mathbf{x}^{\mathrm{obs}}, \mathbf{z})p(\mathbf{z}|\mathbf{x}^{\mathrm{obs}})d\mathbf{z}. \tag{13}$$

Again, this distribution appears out of reach because of the integration of the latent variable $\mathbf{z}$. However, it is reasonable to assume that, for all $\boldsymbol{\eta}$, it is easy to sample from the marginals of $\Phi(\cdot|\boldsymbol{\eta})$. This is for instance the case for Gaussian observation models and factorised observation models (like products of Bernoulli or Poisson distributions). A direct consequence of this assumption is that, for all $\mathbf{z}$, it is easy to sample from $p_{\boldsymbol{\theta}}(\mathbf{x}^{\mathrm{miss}}|\mathbf{x}^{\mathrm{obs}}, \mathbf{z})$. Under this simple assumption, we will see that generating data according to the conditional distribution is actually (asymptotically) possible.

## 3.1 Pseudo-Gibbs sampling

Rezende et al. (2014) proposed a simple way of imputing $\mathbf{x}^{\mathrm{miss}}$ by following a Markov chain $(\mathbf{z}_t, \hat{\mathbf{x}}_t^{\mathrm{miss}})_{t \geq 1}$ (initialised by randomly imputing the missing data with $\hat{\mathbf{x}}_0^{\mathrm{miss}}$). For all $t \geq 1$, the chain alternatively generates $\mathbf{z}_t \sim \Psi(\mathbf{z}|g_{\boldsymbol{\gamma}}(\mathbf{x}^{\mathrm{obs}}, \hat{\mathbf{x}}_{t-1}^{\mathrm{miss}}))$ and $\hat{\mathbf{x}}_t^{\mathrm{miss}} \sim p_{\boldsymbol{\theta}}(\mathbf{x}^{\mathrm{miss}}|\mathbf{x}^{\mathrm{obs}}, \mathbf{z})$ until convergence. This scheme closely resembles Gibbs sampling (Geman and Geman, 1984), and actually exactly coincides with Gibbs sampling when the amortised variational distribution $\Psi(\mathbf{z}|g_{\boldsymbol{\gamma}}(\mathbf{x}^{\mathrm{obs}}, \hat{\mathbf{x}}^{\mathrm{miss}}))$ is equal to the true posterior distribution $p_{\boldsymbol{\theta}}(\mathbf{z}|\mathbf{x}^{\mathrm{obs}}, \hat{\mathbf{x}}^{\mathrm{miss}})$ for all possible $\hat{\mathbf{x}}^{\mathrm{miss}}$. Following the terminology of Heckerman et al. (2000), we will call this algorithm *pseudo-Gibbs sampling*. Very similar schemes have been proposed for more general autoencoder settings (Goodfellow et al., 2016, Section 20.11). Because of its flexibility, this pseudo-Gibbs approach is routinely used for missing data imputation using DLVMs (see e.g. Li et al., 2016; Rezende et al., 2016; Du et al., 2018). Rezende et al. (2014, Proposition F.1) proved that, when these two distributions are close in some sense, pseudo-Gibbs sampling generates points that approximatively follow the conditional distribution $p_{\boldsymbol{\theta}}(\mathbf{x}^{\mathrm{miss}}|\mathbf{x}^{\mathrm{obs}})$. Actually, we will see that a simple modification of this scheme allows to generate *exactly* according to the conditional distribution.

## 3.2 Metropolis-within-Gibbs sampling

At each step of the chain, rather than generating codes according to the approximate posterior distribution, we may *use this approximation as a proposal within a Metropolis-Hastings algorithm* (Metropolis et al., 1953; Hastings, 1970), using the fact that we have access to the unnormalised posterior density of the latent codes.

Specifically, at each step, we will generate a new code $\tilde{\mathbf{z}}_t$ as a proposal using the approximate posterior $\Psi(\mathbf{z}|g_{\boldsymbol{\gamma}}(\mathbf{x}^{\mathrm{obs}}, \hat{\mathbf{x}}_{t-1}^{\mathrm{miss}}))$. This proposal is kept as a valid code with acceptance probability $\rho_t$, defined in Algorithm 1. This probability corre-

---

**Algorithm 1** Metropolis-within-Gibbs sampler for missing data imputation using a trained VAE

---

**Inputs:** Observed data $\mathbf{x}^{\mathrm{obs}}$, trained VAE $(f_{\boldsymbol{\theta}}, g_{\boldsymbol{\gamma}})$, number of iterations $T$
**Outputs:** Markov chain of imputations $\hat{\mathbf{x}}_1^{\mathrm{miss}}, ..., \hat{\mathbf{x}}_T^{\mathrm{miss}}$.
**Initialise** $(\mathbf{z}_0, \hat{\mathbf{x}}_0^{\mathrm{miss}})$
**for** $t = 1$ **to** $T$ **do**
$\quad \tilde{\mathbf{z}}_t \sim \Psi(\mathbf{z}|g_{\boldsymbol{\gamma}}(\mathbf{x}^{\mathrm{obs}}, \hat{\mathbf{x}}_{t-1}^{\mathrm{miss}}))$
$\quad \tilde{\rho}_t = \dfrac{\Phi(\mathbf{x}^{\mathrm{obs}}, \hat{\mathbf{x}}_{t-1}^{\mathrm{miss}}|f_{\boldsymbol{\theta}}(\tilde{\mathbf{z}}_t))p(\tilde{\mathbf{z}}_t)}{\Phi(\mathbf{x}^{\mathrm{obs}}, \hat{\mathbf{x}}_{t-1}^{\mathrm{miss}}|f_{\boldsymbol{\theta}}(\mathbf{z}_{t-1}))p(\mathbf{z}_{t-1})} \dfrac{\Psi(\mathbf{z}_{t-1}|g_{\boldsymbol{\gamma}}(\mathbf{x}^{\mathrm{obs}}, \hat{\mathbf{x}}_{t-1}^{\mathrm{miss}}))}{\Psi(\tilde{\mathbf{z}}_t|g_{\boldsymbol{\gamma}}(\mathbf{x}^{\mathrm{obs}}, \hat{\mathbf{x}}_{t-1}^{\mathrm{miss}}))}$
$\quad \rho_t = \min\{\tilde{\rho}_t, 1\}$
$\quad \mathbf{z}_t = \begin{cases} \tilde{\mathbf{z}}_t & \text{with probability } \rho_t \\ \mathbf{z}_{t-1} & \text{with probability } 1 - \rho_t \end{cases}$
$\quad \hat{\mathbf{x}}_t^{\mathrm{miss}} \sim p_{\boldsymbol{\theta}}(\mathbf{x}^{\mathrm{miss}}|\mathbf{x}^{\mathrm{obs}}, \mathbf{z}_t)$
**end for**

---

sponds to a *ratio of importance ratios*, and is equal to one when the posterior approximation is perfect. This code-generating scheme exactly corresponds to performing a single iteration of an *independent Metropolis-Hastings algorithm*. With the obtained code $\mathbf{z}_t$, we can now generate a new imputation using the exact conditional $\Phi(\mathbf{x}^{\mathrm{miss}}|\mathbf{x}^{\mathrm{obs}}, f_{\boldsymbol{\theta}}(\mathbf{z}_t))$. The obtained algorithm, detailed in Algorithm 1, is a particular instance of a Metropolis-within-Gibbs algorithm. Actually, it exactly corresponds to

the algorithm described by Gelman (1993, Section 4.4), and is *ensured to asymptotically produce samples from the true conditional distribution* $p_{\boldsymbol{\theta}}(\mathbf{x}^{\text{miss}}|\mathbf{x}^{\text{obs}})$, even if the variational approximation is imperfect. Note that when the variational approximation is perfect, all proposals are accepted and the algorithm exactly reduces to Gibbs sampling.

The theoretical superiority of the Metropolis-within-Gibbs scheme compared to the pseudo-Gibbs sampler comes *with almost no additional computational cost*. Indeed, all the quantities that need to be computed in order to compute the acceptance probability need also to be computed within the pseudo-Gibbs scheme—except for prior evaluations, which are assumed to be computationally negligible. However, a poor initialisation of the missing values might lead to a lot of rejections at the beginning of the chain, and to slow convergence. A good initialisation heuristic is to perform a few pseudo-Gibbs iterations at first in order to begin with a sensible imputation. Note also that, similarly to the pseudo-Gibbs sampler, our Metropolis-within-Gibbs scheme can be extended to many other variational approximations—like normalising flows (Rezende and Mohamed, 2015; Kingma et al., 2016)—in a straightforward manner.

## 4 Empirical results

In this section, we investigate the empirical realisations of our theoretical findings on DLVMs. For architecture and implementation details, see Appendix E (in the supplementary material).

### 4.1 Witnessing likelihood blow-up

To investigate if the unboundedness of the likelihood of a DLVM with a Gaussian observation model has actual concrete consequences for variational inference, we train two DLVMs on the Frey faces data set: one with no constraints, and one with the constraint of Proposition 2 (with $\xi = 2^{-4}$). The results are presented in Fig. 2. One can notice that the unconstrained DLVM finds models with very high likelihood but very poor generalisation performance. This confirms that the unboundedness of the likelihood is not a merely theoretical concern. We also display the two upper bounds of the likelihood. The nonparametric bound offers a slight but significant improvement over the naive upper bound. On this example, using the

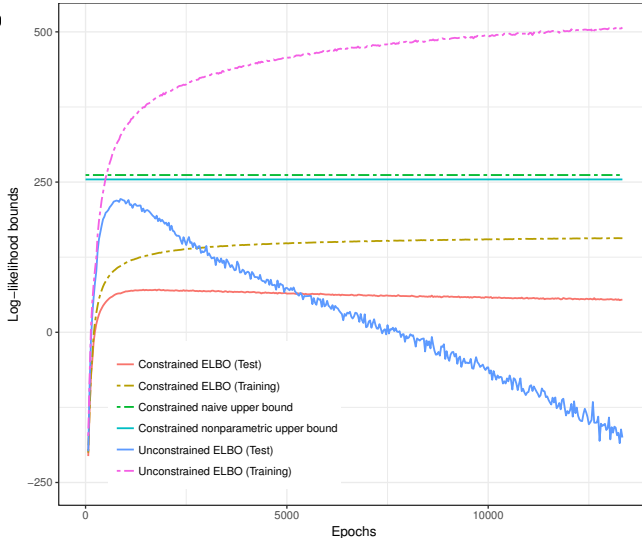

Figure 2: Likelihood blow-up for the Frey Faces data. The unconstrained ELBO appears to diverge, while finding increasingly poor models.

nonparametric upper bound as an early stopping criterion for the unconstrained ELBO appears to provide a good regularisation scheme—that perform better than the covariance constraints on this data set. This illustrates the potential practical usefulness of the connection that we drew between DLVMs and nonparametric mixtures.

### 4.2 Comparing the pseudo-Gibbs and Metropolis-within-Gibbs samplers

We compare the two samplers for single imputation of the test sets of three data sets: Caltech 101 Silhouettes and statically binarised versions of MNIST and OMNIGLOT. We consider two missing data scenarios: a first one with pixels missing uniformly at random (the fractions of missing data considered are $40\%, 50\%, 60\%, 70\%$, and $80\%$) and one where the top or bottom half of the pixels was removed. Both samplers use the same trained VAE and perform the same number of iterations. The imputations are made by computing the means of the chains, which estimate the conditional expected value of the missing data. Since the imputation of these high-dimensional binary data sets can be interpreted as imbalanced binary classification problems, we use the F1 score (the harmonic

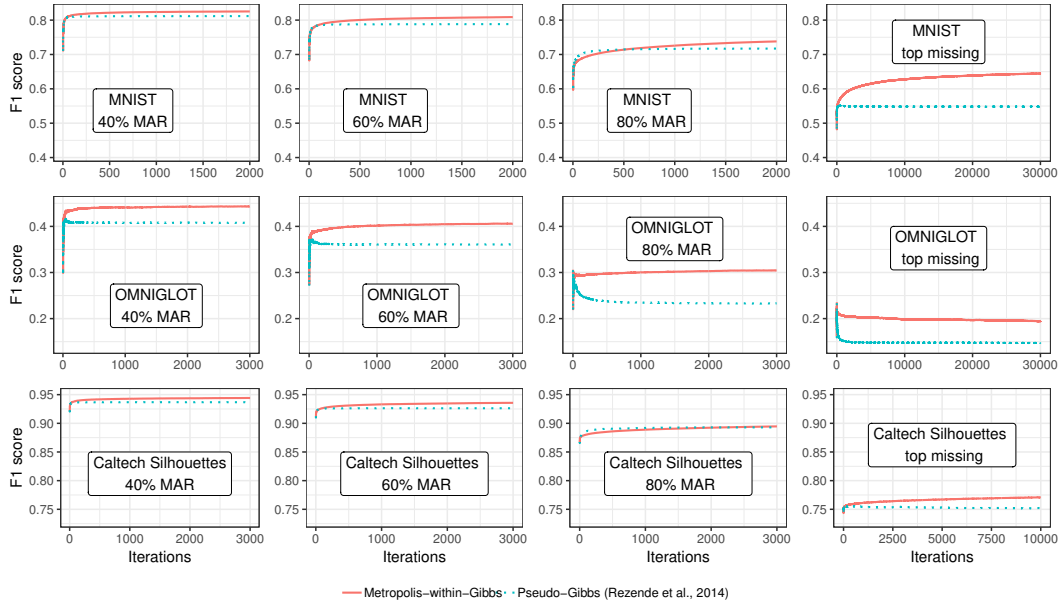

Figure 3: Single imputation results (F1 score between the true and imputed values) for the two Markov chains. Additional results for the bottom missing and the $50\%$ and $70\%$ MAR cases are provided as supplementary material. The more the conditional distribution is challenging (high-dimensional in the MAR cases and highly multimodal in the top/bottom cases), the more the performance gain of our Metropolis-within-Gibbs scheme is important.

mean of precision and recall) as a performance metric. For both schemes, we use 50 iterations of Pseudo-Gibbs as burn-in. In practice, convergence and mixing of the chains can be monitored using a validation set of complete data. The results are displayed on Fig. 3 and in Appendix F (see supplementary material). The chains converge much faster for the missing at random (MAR) situation than for the top/bottom missing scenario. This is probably due to the fact that the conditional distribution of the missing half of an image is highly multimodal. The Metropolis-within-Gibbs sampler consistently outperforms the pseudo-Gibbs scheme, especially for the most challenging scenarios where the top/bottom of the image is missing. One can see that the pseudo-Gibbs sampler appears to converge quickly to a stationary distribution that gives suboptimal results. Because of the rejections, the Metropolis-within-Gibbs algorithm converges slower, but to a much more accurate conditional distribution.

## 5    Conclusion

Although extremely difficult to compute in practice, the exact likelihood of DLVMs offers several important insights on deep generative modelling. An important research direction for future work is the design of principled regularisation schemes for maximum likelihood estimation.

The objective evaluation of deep generative models remains an open question. Missing data imputation is often used as a performance metric for DLVMs (e.g. Li et al., 2016; Du et al., 2018). Since both algorithms have essentially the same computational cost, this motivates to replace pseudo-Gibbs sampling by Metropolis-within-Gibbs when evaluating these models. Upon convergence, the samples generated by Metropolis-within-Gibbs do not depend on the inference network, and explicitly depend on the prior, which allows us to evaluate mainly the generative performance of the models.

We interpreted DLVMs as parsimonious submodels of nonparametric mixture models. While we used this connection to provide upper bounds of the likelihood, many other applications could be derived. In particular, the important body of work regarding consistency of maximum likelihood estimates for nonparametric mixtures (e.g. Kiefer and Wolfowitz, 1956; van de Geer, 2003; Chen, 2017) could be leveraged to study the asymptotics of DLVMs.

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
