[Supplementary Material]

# Leveraging the Exact Likelihood of Deep Latent Variable Models – *Appendices*

**Pierre-Alexandre Mattei**
Department of Computer Science
IT University of Copenhagen
pima@itu.dk

**Jes Frellsen**
Department of Computer Science
IT University of Copenhagen
jefr@itu.dk

## Appendix A. Proof of Theorem 1

**Theorem 1.** *For all $i^* \in \{1, ..., n\}$ and $\mathbf{w} \in \mathbb{R}^d \setminus \{0\}$, we have $\lim_{k \to +\infty} \ell\left(\boldsymbol{\theta}_k^{(i^*, \mathbf{w})}\right) = +\infty$.*

*Proof.* Let $i^* \in \{1, ..., n\}$ and $\mathbf{w} \in \mathbb{R}^d \setminus \{0\}$. To avoid notational overload, we denote $\boldsymbol{\theta}_k = \boldsymbol{\theta}_k^{(i^*, \mathbf{w})}$ in the remainder of this proof. We will show that the contribution $\log p_{\boldsymbol{\theta}_k}(\mathbf{x}_{i^*})$ of the $i^*$-th observation explodes while all other contributions remain bounded below.

A first useful remark is the fact that, since $\mathbf{z}$ is continuous and has zero mean and $\mathbf{w} \in \mathbb{R}^d \setminus \{0\}$, the univariate random variable $\mathbf{w}^T \mathbf{z}$ is continuous and has zero mean. Therefore $\mathbb{P}(\mathbf{w}^T \mathbf{z} \leq 0) > 0$ and $\mathbb{P}(\mathbf{w}^T \mathbf{z} \geq 0) > 0$.

Regarding the $i^*$-th observation, we have for all $k \in \mathbb{N}^*$,

$$p_{\boldsymbol{\theta}_k}(\mathbf{x}_{i^*}) = \int_{\mathbb{R}^d} \mathcal{N}(\mathbf{x}_{i^*} | \mathbf{x}_{i^*}, \boldsymbol{\Sigma}_{\boldsymbol{\theta}_k}(\mathbf{z})) p(\mathbf{z}) d\mathbf{z} \tag{1}$$

$$\geq \int_{\mathbf{w}^T \mathbf{z} \leq 0} \mathcal{N}(\mathbf{x}_{i^*} | \mathbf{x}_i, \boldsymbol{\Sigma}_{\boldsymbol{\theta}_k}(\mathbf{z})) p(\mathbf{z}) d\mathbf{z} \tag{2}$$

$$\geq \int_{\mathbf{w}^T \mathbf{z} \leq 0} |2\pi \boldsymbol{\Sigma}_{\boldsymbol{\theta}_k}(\mathbf{z})|^{-1/2} p(\mathbf{z}) d\mathbf{z}. \tag{3}$$

Let $\mathbf{z} \in \mathbb{R}^d$ such that $\mathbf{w}^T \mathbf{z} \leq 0$. The function

$$\varphi : \alpha \mapsto \alpha \tanh\left(\alpha \mathbf{w}^T \mathbf{z}\right) - \alpha,$$

is strictly decreasing on $\mathbb{R}^+$. Indeed, its derivative is equal to

$$\varphi'(\alpha) = \tanh(\alpha \mathbf{w}^T \mathbf{z}) - 1 + (1 - \tanh^2(\alpha \mathbf{w}^T \mathbf{z})) \alpha \mathbf{w}^T, \tag{4}$$

which is strictly negative because the image of the hyperbolic tangent function is $]-1, 1[$. Moreover,

$$\lim_{\alpha \to \infty} \alpha \tanh\left(\alpha \mathbf{w}^T \mathbf{z}\right) - \alpha = -\infty.$$

Therefore, the sequence $(|2\pi \boldsymbol{\Sigma}_{\boldsymbol{\theta}_k}(\mathbf{z})|^{-1/2})_{k \geq 1}$ is strictly increasing and diverges to $+\infty$ for all $\mathbf{z} \in \mathbb{R}^d$ such that $\mathbf{w}^T \mathbf{z} \leq 0$. Therefore , the monotone convergence theorem combined with the fact that $\mathbb{P}(\mathbf{w}^T \mathbf{z} \leq 0) > 0$ insure that the right hand side of (1) diverges to $+\infty$, leading to $p(\mathbf{x}_{i^*} | \boldsymbol{\mu}_{\boldsymbol{\theta} i^*}, \boldsymbol{\Sigma}_{\boldsymbol{\theta}_k}) \to +\infty$.

Regarding the other contributions, let $j \neq i^*$. Since $\mathbb{P}(\mathbf{w}^T\mathbf{z} > 0) > 0$, there exists $\varepsilon > 0$ such that $\mathbb{P}(\mathbf{w}^T\mathbf{z} \geq \varepsilon) > 0$. We have

$$p_{\boldsymbol{\theta}_k}(\mathbf{x}_j) \geq \int_{\mathbf{w}^T\mathbf{z} \geq \varepsilon} \mathcal{N}(\mathbf{x}_j | \mathbf{x}_{i^*}, \boldsymbol{\Sigma}_{\boldsymbol{\theta}_k}(\mathbf{z}))p(\mathbf{z})d\mathbf{z}$$

$$= \int_{\mathbf{w}^T\mathbf{z} \geq \varepsilon} \frac{\exp\left(\frac{-||\mathbf{x}_j - \mathbf{x}_{i^*}||_2^2}{2\exp(\alpha_k \tanh(\alpha_k \mathbf{w}^T\mathbf{z}) - \alpha_k)}\right)}{(2\pi \exp(\alpha_k \tanh(\alpha_k \mathbf{w}^T\mathbf{z}) - \alpha_k))^{p/2}} p(\mathbf{z})d\mathbf{z}$$

$$= \frac{1}{(2\pi)^{p/2}} \int_{\mathbf{w}^T\mathbf{z} \geq \varepsilon} \exp\left(\frac{-||\mathbf{x}_j - \mathbf{x}_{i^*}||_2^2}{2\exp(\alpha_k \tanh(\alpha_k \mathbf{w}^T\mathbf{z}) - \alpha_k)}\right) p(\mathbf{z})d\mathbf{z}$$

$$\geq \frac{1}{(2\pi)^{p/2}} \exp\left(\frac{-||\mathbf{x}_j - \mathbf{x}_{i^*}||_2^2}{2\exp(\alpha_k \tanh(\alpha_k \varepsilon) - \alpha_k)}\right) \mathbb{P}(\mathbf{w}^T\mathbf{z} \geq \varepsilon),$$

and, since $\lim_{\alpha \to \infty} \alpha \tanh(\alpha\varepsilon) - \alpha = 0$, we will have

$$\liminf_{k \to \infty} p_{\boldsymbol{\theta}_k}(\mathbf{x}_j) \geq \frac{\mathbb{P}(\mathbf{w}^T\mathbf{z} \geq \varepsilon)}{(2\pi)^{p/2}} \exp\left(\frac{-||\mathbf{x}_j - \mathbf{x}_{i^*}||_2^2}{2}\right),$$

therefore

$$\liminf_{k \to \infty} p_{\boldsymbol{\theta}_k}(\mathbf{x}_j) > 0.$$

By combining all contributions, we end up with $\lim_{k \to +\infty} \ell(\boldsymbol{\theta}_k) = +\infty$. $\qquad \square$

## Appendix B. Proof of Proposition 1

**Proposition 1.** *For all $k \in \mathbb{N}^*$, $i^* \in \{1, ..., n\}$, and $\mathbf{w} \in \mathbb{R}^d \setminus \{0\}$, the distribution $p_{\boldsymbol{\theta}_k^{(i^*,\mathbf{w})}}$ is spherically symmetric and unimodal around $\mathbf{x}_{i^*}$.*

*Proof.* Let $k \in \mathbb{N}^*$, $i^* \in \{1, ..., n\}$, and $\mathbf{w} \in \mathbb{R}^d \setminus \{0\}$. To avoid notational overload, we denote $\boldsymbol{\theta}_k = \boldsymbol{\theta}_k^{(i^*,\mathbf{w})}$. We have, for all $\mathbf{x} \in \mathbb{R}^p$,

$$p_{\boldsymbol{\theta}_k^{(i^*,\mathbf{w})}}(\mathbf{x}) = \int_{\mathbb{R}^d} \mathcal{N}(\mathbf{x}|\mathbf{x}_{i^*}, \boldsymbol{\Sigma}_{\boldsymbol{\theta}_k}(\mathbf{z}))p(\mathbf{z})d\mathbf{z} = \int_{\mathbb{R}^d} \mathcal{N}(\mathbf{x} - \mathbf{x}_{i^*}|\mathbf{0}_p, \boldsymbol{\Sigma}_{\boldsymbol{\theta}_k}(\mathbf{z})), p(\mathbf{z})d\mathbf{z}.$$

The density of $\mathbf{x}$ is therefore a decreasing function of $||\mathbf{x} - \mathbf{x}_i||_2$, hence the spherical symmetry and the unimodality. Note that there are several different definitions for multivariate unimodality (see e.g. Dharmadhikari and Joag-Dev, 1988). Here, we mean that the only local maximum of the density is at $\mathbf{x}_i$. $\qquad \square$

## Appendix C. Proof of Theorem 2

**Theorem 2.** *Assume that $(\Phi(\cdot|\boldsymbol{\eta}))_{\boldsymbol{\eta} \in H}$ is the family of multivariate Bernoulli distributions or the family of Gaussian distributions with the spectral constraint of Proposition 2. The likelihood of the corresponding nonparametric mixture model is maximised for a finite mixture model of $k \leq n$ distributions from the family $(\Phi(\cdot|\boldsymbol{\eta}))_{\boldsymbol{\eta} \in H}$.*

*Proof.* Let us assume that there are $L \leq n$ distinct observations $\tilde{\mathbf{x}}_1, ..., \tilde{\mathbf{x}}_L$. Following Lindsay (1983), let us consider the *trace of the likelihood curve*

$$\Gamma = \{(\Phi(\tilde{\mathbf{x}}_l|\boldsymbol{\eta}))_{l \leq L} \,|\, \boldsymbol{\eta} \in H\}.$$

We will use the following theorem, which describes maximum likelihood estimators of nonparametric mixtures under some topological conditions.

**Theorem (Lindsay, 1983, Theorem 3.1).** *If $\Gamma$ is compact, then the likelihood of the corresponding nonparametric mixture model is maximised for a finite mixture model of $k \leq n$ distributions from the family $(\Phi(\cdot|\boldsymbol{\eta}))_{\boldsymbol{\eta} \in H}$.*

This theorem allows us to treat both cases:

**Bernoulli observation model.** Since $H = [0, 1]^p$ is compact and the function $\boldsymbol{\eta} \mapsto (\Phi(\tilde{\mathbf{x}}_l|\boldsymbol{\eta}))_{l \leq L}$ is continuous, $\Gamma$ is compact and Lindsay's theorem can be directly applied.

**Gaussian observation model.** The parameter space $H$ is not compact in this case, but we can get around this problem using a compactification argument similar to the one of van der Vaart and Wellner (1992). Consider the Alexandroff compactification $H \cup \{\infty\}$ of the parameter space (Kelley, 1955, p. 150). Because of the definition of $H$, we have $\lim_{\boldsymbol{\eta} \to \infty} \Phi(\tilde{\mathbf{x}}_l|\boldsymbol{\eta}) = 0$ for all $l \in \{1, ..., L\}$. Therefore, we can continuously extend the function $\boldsymbol{\eta} \mapsto (\Phi(\tilde{\mathbf{x}}_l|\boldsymbol{\eta})))_{l \leq L}$ from $H$ to $H \cup \{\infty\}$ using the conventions $\Phi(\tilde{\mathbf{x}}_l|\infty) = 0$ for all $l \in \{1, ..., L\}$. The space $\{(\Phi(\tilde{\mathbf{x}}_l|\boldsymbol{\eta}))_{l \leq L} \,|\, \boldsymbol{\eta} \in H \cup \{\infty\}\}$ is therefore compact, and we can use Lindsay's theorem to deduce that the nonparametric maximum likelihood estimator for the compactified parameter space $H \cup \{\infty\}$ is a finite mixture of distributions from $(\Phi(\cdot|\boldsymbol{\eta}))_{\boldsymbol{\eta} \in H \cup \{\infty\}}$. However, this finite mixture can only contain elements from $(\Phi(\cdot|\boldsymbol{\eta}))_{\boldsymbol{\eta} \in H}$. Indeed, if any mixture component were associated with $\boldsymbol{\eta} = \infty$, the likelihood could be improved by emptying said component. Therefore, the maximum likelihood estimator found using the compactified space is also the maximum likelihood estimator of the original space, which allows us to conclude. $\square$

## Appendix D. Proof of Theorem 3

**Theorem 3.** *Assume that*

1. *$(\Phi(\cdot|\boldsymbol{\eta}))_{\boldsymbol{\eta} \in H}$ is the family of multivariate Bernoulli distributions or the family of Gaussian distributions with the spectral constraint of Proposition 2.*

2. *The decoder has universal approximation abilities : for any compact $C \subset \mathbb{R}^d$ and continuous function $f : C \to H$, for all $\varepsilon > 0$, there exists $\boldsymbol{\theta}$ such that $||f - f_{\boldsymbol{\theta}}||_\infty < \varepsilon$.*[1]

*Then, for all $\varepsilon > 0$, there exists $\boldsymbol{\theta}$ such that*

$$\ell(\hat{G}) \geq \ell(\boldsymbol{\theta}) \geq \ell(\hat{G}) - \varepsilon.$$

*Proof.* The left hand side of the inequality directly comes from the first assumption and Theorem 2. We will now prove the right hand side.

Let $\varepsilon > 0$. Since the logarithm is continuous, there exists $\delta > 0$ such that

$$0 \leq u \leq \delta \implies \sum_{i=1}^{n} \log(p_{\hat{G}}(\mathbf{x}_i) - u) \geq \ell(\hat{G}) - \varepsilon. \tag{5}$$

We will show that $p_{\hat{G}}(\mathbf{x}_i)$ and $p_{\boldsymbol{\theta}}(\mathbf{x}_i)$ can be made closer than $\delta$ for all $i \leq n$.

Theorem 2 insures that $\hat{G}$ is a finite mixture, so there exist some $(\pi_1, ..., \pi_K) \in \Delta_K$ and $\boldsymbol{\eta}_1, ..., \boldsymbol{\eta}_K \in H$ such that

$$\forall \mathbf{x} \in \mathcal{X}, \; p_{\hat{G}}(\mathbf{x}) = \sum_{k=1}^{K} \pi_k \Phi(\mathbf{x}|\boldsymbol{\eta}_k).$$

We will partition the domain of integration of the latent variable into several parts:

- an infinite part of very small prior mass,

- a compact set of high prior mass over which we can apply the universal approximation property.

The compact set will be divided itself into $2K$ parts: one part for each mixture component and $K - 1$ small parts to insure the continuity of the decoder. More specifically, let $e \in ]0, \min\{\pi_1, ..., \pi_K\}[$, let $F$ be the (prior) cumulative distribution function of $||\mathbf{z}||_2$, and let:

- $\alpha_1 = 0$,

- $\forall k \in \{2, ..., K\},\ \alpha_k = F^{-1}(\sum_{l=1}^{k-1} \pi_l)$,

- $\alpha_{k+1} = +\infty$,

- $\forall k \in \{1, ..., K\},\ \beta_k = F^{-1}(\sum_{l=1}^{k} \pi_l - e)$.

Let $C = \{\mathbf{z} \in \mathbb{R}^d \mid ||\mathbf{z}||_2 \leq \beta_K\}$. Consider a continuous function $f : C \to H$ such that

$$
\forall \mathbf{z} \in K, f(\mathbf{z}) =
\begin{cases}
\boldsymbol{\eta}_1 & \text{if } \alpha_1 \leq ||\mathbf{z}||_2 \leq \beta_1 \\
\boldsymbol{\eta}_2 & \text{if } \alpha_2 \leq ||\mathbf{z}||_2 \leq \beta_2 \\
... & \\
\boldsymbol{\eta}_K & \text{if } \alpha_K \leq ||\mathbf{z}||_2 \leq \beta_K.
\end{cases}
$$

For example, $f$ can be built using the Tietze extension theorem (see e.g. Kelley, 1955, p. 242) together with the conditions of the above formula. According to the universal approximation property, we can find decoders arbitrarily close to $f$. How close do they need to be? Invoking the continuity of the functions $(\boldsymbol{\eta} \mapsto \Phi(\mathbf{x}_i|\boldsymbol{\eta}))_{i \leq n}$ in $\boldsymbol{\eta}_1, ..., \boldsymbol{\eta}_K$, there exists $\Delta > 0$ such that

$$
\forall l \in \{1, ..., K\},\ ||\boldsymbol{\eta}_k - \boldsymbol{\eta}||_\infty \leq \Delta \implies |\Phi(\mathbf{x}_i|\boldsymbol{\eta}) - \Phi(\mathbf{x}_i|\boldsymbol{\eta}_k)| \leq e.
$$

We will consider now a decoder $f_{\boldsymbol{\theta}}$ such that $||f - f_{\boldsymbol{\theta}}||_\infty \leq \Delta$.

We can write, for all $i \leq n$,

$$
\begin{aligned}
p_{\boldsymbol{\theta}}(\mathbf{x}_i) &= \sum_{l=1}^{K} \left( \int_{\alpha_l \leq ||\mathbf{z}||_2 \leq \beta_l} \Phi(\mathbf{x}_i|f_{\boldsymbol{\theta}}(\mathbf{z}))p(\mathbf{z})d\mathbf{z} + \int_{\beta_l \leq ||\mathbf{z}||_2 \leq \alpha_{l+1}} \Phi(\mathbf{x}_i|f_{\boldsymbol{\theta}}(\mathbf{z}))p(\mathbf{z})d\mathbf{z} \right) \\
&\geq \sum_{l=1}^{K} \int_{\alpha_l \leq ||\mathbf{z}||_2 \leq \beta_l} \Phi(\mathbf{x}_i|f_{\boldsymbol{\theta}}(\mathbf{z}))p(\mathbf{z})d\mathbf{z}.
\end{aligned}
$$

Using the fact that $||f - f_{\boldsymbol{\theta}}||_\infty \leq \Delta$, we have therefore, for all $i \leq n$,

$$
\begin{aligned}
p_{\boldsymbol{\theta}}(\mathbf{x}_i) &\geq \sum_{l=1}^{K} \int_{\alpha_l \leq ||\mathbf{z}||_2 \leq \beta_l} (\Phi(\mathbf{x}_i|\boldsymbol{\eta}_l) - e)p(\mathbf{z})d\mathbf{z} \\
&= \sum_{l=1}^{K} (\Phi(\mathbf{x}_i|\boldsymbol{\eta}_l) - e)(\pi_l - e) \\
&= \sum_{l=1}^{K} \left( \pi_l \Phi(\mathbf{x}_i|\boldsymbol{\eta}_l) + e^2 - e(\pi_l + \Phi(\mathbf{x}_i|\boldsymbol{\eta}_l)) \right) \\
&\geq \sum_{l=1}^{K} \pi_l \Phi(\mathbf{x}_i|\boldsymbol{\eta}_l) - e \left( \sum_{l=1}^{K} \Phi(\mathbf{x}_i|\boldsymbol{\eta}_l) + 1 \right).
\end{aligned}
$$

Now, if we take $e$ such that

$$
e \left( \sum_{l=1}^{K} \Phi(\mathbf{x}_i|\boldsymbol{\eta}_l) + 1 \right) \leq \delta,
$$

for all $i \leq n$, we end up with

$$
p_{\boldsymbol{\theta}}(\mathbf{x}_i) \geq p_{\hat{G}}(\mathbf{x}_i) - \delta.
$$

Using (5), this eventually leads to $\ell(\boldsymbol{\theta}) \geq \ell(\hat{G}) - \varepsilon$. $\qquad \square$

## Appendix E. Implementation details

Some computational choices are common for all experiments: the prior distribution is a standard Gaussian distribution; the chosen variational family $(\Psi(\cdot|\boldsymbol{\kappa}))_{\boldsymbol{\kappa}\in K}$ is the family of Gaussian distributions with "diagonal + rank-1" covariance (as in Rezende et al., 2014, Section 4.3); stochastic gradients of the ELBO are computed via the *path derivative estimate* of Roeder et al. (2017); the Adam optimiser (Kingma and Ba, 2014) is used with a learning rate of $10^{-4}$ and mini-batches of size 10; neural network are initialised following the heuristics of Glorot and Bengio (2010); sampling for variational autoencoders is performed via the Distributions module of TensorFlow (Dillon et al., 2017).

For the Frey faces data set, we used a Gaussian DLVM together with the parametrisation presented in Section 2.1 (with $d = 5$ and $h = 200$). The architecture of the inference network follows the same parametrisation. Constrained finite Gaussian mixtures were fit using the scikit-learn package (Pedregosa et al., 2011). Regarding the missing data experiments, we used MLPs with ReLU activations and 200 hidden units together with an intrinsic dimension of 50.

## Appendix F. Additional imputation experiments

Figure 1: Single imputation results (F1 score between the true and imputed values) for the two Markov chains.

## Footnotes

[1]Here, the infinite norm is defined as follows. For any $\boldsymbol{\eta} \in H$, $||\boldsymbol{\eta}||_\infty$ is the maximum of the absolute values of the coordinates of $\boldsymbol{\eta}$ (which is well-defined because the observation model is parametric). For any $f : C \to H$, we note then $||f||_\infty = \max_{\mathbf{z} \in C} ||f(\mathbf{z})||_\infty$.