[Reviews · NeurIPS 2018]

Reviewer 1



Updated Review after Rebuttal: After reading the authors response and re-evaluate the paper I do agree that most of my concerns that there was a fundamental issue with some of their statements were wrong, hence I'm changing my score from 3 to 6. From going into detail of the proof it resides on constructing a generative model where for half of the latent variables (w^t z < 0) the integral is bounded for all data points and for the other half for 1 data point the integral diverges while for the other goes to zero. This split allows them to say that the one integral diverges and the all of the others are finite hence the likelihood is infinite. However, I'm still not convinced that this issue actually arises at all in practical settings. First, in practice, we are optimizing an ELBO which is never tight, hence for this to be convincing argument the authors should investigate whether there are settings of the ELBO where it diverges except when it can perfectly reconstruct the posterior. Furthermore, I still stand that I do not think that the results on the Frey Faces dataset are interpreted correctly and given that this is a fairly small dataset it is highly likely that the generative model overfits to the data (but not in the way for the divergence to happen). The experimental section in this direction seems to be a bit weak, nevertheless, the paper is worth being accepted. ================================================================ Summary of paper: The paper investigates various properties of the exact likelihood of Deep Latent Variable Models (DLVM). In the first part of the exposition, the authors argue that the likelihood of standard DLVMs is ill-posed in the sense that there exist parameters for which it diverges for continuous data and is well-posed for discrete. In the second part, they propose to use Metropolis-within-Gibbs instead of the standardly used pseudo-Gibbs sampler for missing data imputation. The experimental section on demonstrating unboundedness of the likelihood is very brief and needs more evidence, while the ones comparing different samplers is much more convincing. Summary of review: The paper's arguments and conclusions about regarding the likelihood being unbounded for DLVMs seem to be fundamentally wrong. Although the proved theorems are mathmeatically correct their assumptions do not match the standard way these models are defined in the literature. Additionally, the experiments in this direction are minimal and I suspect that the reason for the generalization gap is that by not treating the network parameters in a Bayesian way we are still performing MLE on those, hence we are not guaranteed that the network can not overfit. The proposed method for doing missing data imputation - Metropolis-within-Gibbs - is not known to me to have been used previously in the literature, hence there is a valid contribution here. The ideas presented clearly and are very easy to implement. Results to support the theoretical intuition that this method should perform well versus the pseudo-Gibbs most widely used in practice. My general opinion is that the authors should reconsider their stance on the unboundness of the likelihood, based on my details comments below, and focus a lot more on the missing data imputation. The idea seems novel and interesting to me. More results and better presentation in this direction would definitely make the paper worth for publishing. However, given the inaccuracies in the first idea, which takes about half of the paper, at the current stage, I propose this work to be rejected. Comments by sections: On the boundedness of the likelihood of deep latent variable models ====================================================== The authors argue that standard DVMs as the one presented in Kingma and Welling [25] have unbounded loglikelihood. Although Theorem 1 is mathematically correct as well as Proposition 1 there is a major flaw in the logic and conclusions extracted from this. The parameterization considered in these and written out in equation (9) has been presented to be reflecting the model described in equation (8) which is a 2 layer generator network of a standard VAE. However, there is a fundamental mistake here - equation (9) assumes that there is a different set of parameters $\theta_i$ **per datapoint**, specifically by their definition $\theta_i$ contains $x_i$. This is significantly flawed as the standard VAE model of Kingma and Welling as well as any other model derived from this framework always assumed that the parameters are tight together between all data points (e.g. as the authors originally define $\theta$ below equation (8)). Under this wrong assumption, the following Theorem 1 seems very natural as this non-parametric model would just output a delta function around each data point. However, in any VAE in the literature, where this is not the case, the arguments of this section would not apply and the likelihood is most-likely bounded (I'm not aware of any way of proving this rigorously either way). Hence and the conclusions drawn regarding models known in the community are wrong. Towards data-dependent likelihood upper bounds ============================================== In this section, the authors propose an upper bound on the likelihood based on finite mixture models. Theorem 2 is a valid bound, however, the assumptions in Theorem 3 are very unrealistic: 1. It is still under the assumption that the generative model parameters are per-data point. 2. It cites the universal approximation abilities of Neural Networks as a reason why the bound is tight. Although this is technically possible, if we assume 1. is not a flaw, however even we never actually train networks with infinite capacity, hence the bound is in fact arbitrarily (and most likely) very loose. Additionally, the authors fail to cite previous works on upper bounds of the likelihood of DLVMs such as "Sandwiching the marginal likelihood using bidirectional Monte Carlo", "Approximate Bayesian inference with the weighted likelihood bootstrap". Specifically, the Bi-directional Monte Carlo Method can provide increasingly more accurate and tighter upper bounds based on the amount of computing resources provided. Missing data imputation using the exact conditional likelihood ============================================================== The de-facto "standard" way for doing data-imputation with DLVMs is to use what the authors call pseudo-gibbs sampling. This invloves sequentially sampling p(z|x_o,x_m) and p(x_m|z,x_o) where one replaces the true posterior with the learned approximation. The authors propose to "fix" the method by rather than just sampling from the approximate posterior to use Metropolis-Hastings to account for the approximation gap. This is as correctly pointed out by the authors an instantion of Metropolis-within-Gibbs and indeed is a very valid observation which I have not seen so far in the literature. Although simple, such connections are indeed important. Witnessing likelihood blow-up ============================================================== Here I think that the experiments are not enough - the authors only demonstrate work on the Frey faces. This is both relatively small as well as not too popular in order to compare with other results in the literature. Additionally, the authors conclude that the big generalization gap is a confirmation of the unboundedness of the likelihood. As discussed earlier, the likelihood is most likely bounded. Secondly, the most likely reason for the generalization gap is the fact that the training on the ELBO still performs an MLE estimation of the parameters $\theta$ of the generative network. This implies that if the network has enough capacity it **will** overfit the data and has been observed before in the literature. This is indeed an issue with the naive formulation of Variational Inference where one does not also provide prior for the network parameters. However, for more complicated datasets, it seems that in practice generating good images is difficult enough that the models used tend to not have enough capacity to do so. Thus my suspicion is that the Frey faces is too simple. Comparing the pseudo-Gibbs and Metropolis-within-Gibbs samplers ============================================================== The results in this section provide good evidence that the Metropolis-within-Gibbs sampler performs bettern tha the pseudo-Gibbs. It would be interesting to show more statistics than just the F1-score (e.g. the raw accuracy etc...). Further, showing actual reconstructed images would make the presentation even better.

Reviewer 2



*Update after authors response* Thank you for clarifying a few points. This paper makes an interesting point and is well written, a good submission in my opinion. Further experiments to look at the likelihood blow-up in practice and quantify the extend of the problem (how often, under which conditions, ...) would make the paper even stronger, this is the main reason why I keep my score at 7. This paper presents three main contributions. First, the authors demonstrate that the likelihood of Deep Latent Variable Model (DLVM) with Gaussian latent variables can be unbounded by providing an example of parameter choices leading to a likelihood blow-up. They further show this cannot happen for Bernoulli latents and that a spectral constraint on the covariance can prevent the issue in the Gaussian case. Second, an upper bound on the likelihood is derived using the fact that DLVM are particular cases of mixture models, the proposed bound is estimated using EM for the corresponding nonparametric mixture model. Third, a metropolis-within-Gibbs exact sampling is proposed to replace the pseudo-Gibbs scheme performed to infer missing input features in DLVM. The analysis of maximum likelihood performed for Gaussian and Bernoulli DLVM provide some interesting and potentially actionable insights. It would be very interesting to see more experiments to study the impact of a non-bounded likelihood in Gaussian DLVM, in particular how often this becomes an issue in practice. While the reasoning linking unboundedness to mode collapse seems sensible it remains unclear if cases of mode collapse experienced in practice are due to this issue. The experimental section presents only one example of likelihood blow-up and further experiments would make the paper stronger. The practical interest of the upper bound provided seems limited by the fact that EM estimation on the corresponding mixture model is needed. Experiments performed with missing data seem more compelling and show a clear improvement on different datasets. About Proposition 2. Does the particular form of the bound provided hold with the assumption that z ~ N(0, I) ? The proof seems to make that assumption. In the proof of Theorem 2. Could the authors explain more precisely how the assumption of a spectral constraint for the covariance is used in the proof of the Gaussian case? Is it needed to obtain the zero limit of the density when eta goes to infinity? About 4.1 We see in figure 2 a collapse of the test log-likelihood for the unconstrained model. How common is this phenomenon, does it depend on the initialisation or the learning rate? It would be interesting to see which values of the parameters correspond to this blow-up scenario and to see if there is indeed a mode collapse in this case. The gap between naive and nonparametric bounds seems relatively small, do the author have any insight on that?

Reviewer 3



MAIN IDEAS The paper shows that the likelihood for a deep latent variable model, p(x | z) p(z), can be made arbitrarily big if both the mean and variance of p(x | z) is learned, and p(x | z) is Gaussian. Here, unlike conventional Bayesian settings, no prior is placed on the parameters of p(x | z); they remain unregularized too. The proof construction relies on an argument that one could construct a p(x | z) that would spike with zero variance at one particular data point in the training set, and have bounded variance elsewhere. It therefore shows that the maximum likelihood estimator is ill-posed. As an aside, it would have been nice if the paper cleanly defined "ill-posed" and maybe linked it to Tikhonov and Arsenin's "Solutions of Ill-Posed Problems"... The paper proceeds to give a simple (spectral contraints) condition on the covariance of p(x | z) to ensure the likelihood is always bounded. Some of the ideas -- that a Bernoulli observation model has bounded likelihood; that its log likelihood is always negative -- is well known. There are two more contributions: 1. We can choose p(x | z) to be more or less flexible, and its capacity limit is characterized by a "parsimony gap". The supplementary material gives very nice proofs (note: the paper doesn't always state that the proofs are in the supplementary material.) 2. An MCMC method for doing approximate missing data imputation. A Metropolis Hastigs accept-reject is added when sampling between x_missing and latent z. (authors check typo: should \rho_t = ... contain a min(..., 1) in Algorithm 1?) STRENGTHS The handy take-home message from this paper is that the Gaussian variance for a VAE has to be constrained, or practitioners risk their models "perfectly modelling" one datapoint in the training set, with zero predictive variance. This overfitting nicely illustrated in Figure 1. It is also immensely interesting to see how the results from finite mixture models (a mixture component describing only one data point, with zero variance on the mixture component) translates to the VAE / deep generative setting. Furthermore, an improved algorithm is provided for sampling x_missing | x_observed, using a VAE encoder / decoder. WEAKNESSES The paper doesn't have one message. Theorem 3 is not empirically investigated. TYPOS, ETC - Abstract. To state that the papers "draws useful connections" is uninformative, if the abstract doesn't state *what* connections are drawn. - Theorem 1. Is subscript k (overloaded later in Line 178, etc) necessary? It looks like one can simply restate the theorem in terms of alpha -> infinity? - Line 137 -- do the authors confuse VAEs with GANs's mode collapse here? - The discussion around equation (10) is very terse, and not very clearly explained. - Line 205. True posterior over which random variables? - Line 230 deserves an explanation, i.e. why conditioning p(x_missing | x_observed, x) is easily computable. - Figure 3: which Markov chain line is red and blue? Label?